# WDR82-Mediated H3K4me3 Is Associated with Tumor Proliferation and Therapeutic Efficacy in Pediatric High-Grade Gliomas

**DOI:** 10.3390/cancers15133429

**Published:** 2023-06-30

**Authors:** Nitin Wadhwani, Sonali Nayak, Yufen Wang, Rintaro Hashizume, Chunfa Jie, Barbara Mania-Farnell, Charles David James, Guifa Xi, Tadanori Tomita

**Affiliations:** 1Department of Pathology, Ann & Robert H. Lurie Children’s Hospital of Chicago, Northwestern University Feinberg School of Medicine, Chicago, IL 60611, USA; 2Division of Pediatric Neurosurgery, Ann & Robert H. Lurie Children’s Hospital of Chicago, Northwestern University Feinberg School of Medicine, Chicago, IL 60611, USA; 3Department of Radio-oncology, Northwestern University Feinberg School of Medicine, Chicago, IL 60611, USA; 4Department of Pediatrics, Northwestern University Feinberg School of Medicine, Chicago, IL 60611, USA; 5Department of Biochemistry and Nutrition, Des Moines University Medicine and Health Sciences, Des Moines, IA 50312, USA; 6Department of Biological Sciences, Purdue University Northwest, Hammond, IN 46323, USA; 7Department of Neurological Surgery, Northwestern University Feinberg School of Medicine, Chicago, IL 60611, USA

**Keywords:** WDR82, epigenetics, proliferation, DNA damage repair, children, high-grade glioma

## Abstract

**Simple Summary:**

Pediatric high-grade gliomas bearing either an H3G34V/R or H3K27M mutation are incurable brain tumors with unique epigenomes. Current epigenomic studies on pHGGs primarily focus on these mutations; however, the function of other important crosstalk histone posttranslational modifications must be determined to elucidate tumor mechanisms. Here, we show that WDR82-mediated H3K4me3 is an important determinant of pediatric glioma malignancy and therapeutic response, and thus a potential epigenetic therapeutic target for children with malignant gliomas.

**Abstract:**

Pediatric high-grade gliomas (pHGGs) are common malignant brain tumors without effective treatment and poor patient survival. Abnormal posttranslational modification at the histone H3 tail plays critical roles in tumor cell malignancy. We have previously shown that the trimethylation of lysine 4 at histone H3 (H3K4me3) plays a significant role in pediatric ependymoma malignancy and is associated with tumor therapeutic sensitivity. Here, we show that H3K4me3 and its methyltransferase WDR82 are elevated in pHGGs. A reduction in H3K4me3 by downregulating WDR82 decreases H3K4me3 promoter occupancy and the expression of genes associated with stem cell features, cell proliferation, the cell cycle, and DNA damage repair. A reduction in WDR82-mediated H3K4me3 increases the response of pediatric glioma cells to chemotherapy. These findings suggest that WDR82-mediated H3K4me3 is an important determinant of pediatric glioma malignancy and therapeutic response. This highlights the need for a more thorough understanding of the potential of WDR82 as an epigenetic target to increase therapeutic efficacy and improve the prognosis for children with malignant gliomas.

## 1. Introduction

Brain tumors, the most common form of solid tumor in children under the age of 15, are responsible for approximately 20% of all childhood cancers. Five-year survival, following diagnosis and treatment of a primary malignant brain tumor, is ~30%. Among pediatric brain tumors, high-grade gliomas (pHGGs), including diffuse intrinsic pontine gliomas (DIPGs), are especially devastating, with an average patient survival of less than 15 months [1]. In the last decade, pHGGs have been extensively characterized and remarkable progress has been made in understanding the mechanisms associated with the tumors at the molecular level.

Mutations at the histone H3 tail, which change the glycine at residue 34 to either valine or arginine (H3G34V/R) in hemisphere tumors, or the lysine to methionine substitution at residue 27 (H3K27M) in DIPGs [2,3], are common. These mutations remodel chromatin and alter posttranslational histone modifications (PTMs), consequently bringing about abnormal gene expression. Typically, H3G34V/R reduces global H3K36me3 through the inhibition of its methyltransferase SETD2 [4]. SETD2 is an antagonist of EZH2, a major functional subunit of polycomb repressive complex 2 (PRC2) [5]. SETD2 inhibition activates PRC2 and increases H3K27 trimethylation (H3K27me3) [6]. H3K27M mutations significantly reduce global H3K27me3 through PRC2 repression [7]. Strikingly, H3K4me3 is nearly unaffected by either histone gene mutation [8,9,10]. This is unusual, given that H3K4me3 and H3K27me3 are important bivalent histone PTMs with respect to the regulation of gene expression in human development and disease [11].

H3K4me3 levels play a role in determining the pathogenesis of various human cancers [12,13], including glial-derived brain tumors [14]. Additionally, H3K4me3 has prognostic utility for multiple cancers [12,15]. High levels of H3K4me3 are associated with the 5′ regions of virtually all active genes, and a strong positive correlation exists between this modification, transcription rates, and active polymerase II occupancy, which is critical for transcriptional activity in a variety of eukaryotic species [16,17]. We have recently shown that high H3K4me3 is associated with poor survival in pediatric ependymoma [14]. Its role in pediatric gliomas has not been characterized.

A human SET domain containing 1A/B protein complexes (hSETD1A/B-COMPASS) binds to DNA containing unmethylated CpG motifs and is responsible for trimethylating H3K4 [18]. Human SETD1A/B proteins exhibit a large non-overlapping subnuclear distribution, indicating their distinct localization at subsets of target genes [19]. These complexes are identical, with the exception of the catalytic component [18,19]. Each complex contains seven units including SETD1A or SETD1B, ASH2, CXXC finger protein 1 (CFP1), DPY30, RBBP5, WDR5, and WDR82 [18,19]. WDR82 is a unique subunit of hSETD1A/B [20]. WDR82 trimethylates H3K4 through recruitment of hSETD1A/B. WDR82 interacts with hSETD1A/B via an N-terminal RNA recognition motif (RRM) within the latter to mark transcription start sites (TSS) of active genes [17,20,21]. During early embryonic development, WDR82 is crucial for H3K4me3 at the promoter of Oct4, whose expression is associated with embryonic stem cells [22]. WDR82 is also a component of the PTW/PP1 phosphatase complex, which is involved in the control of the chromatin structure and cell-cycle progression during the transition from mitosis to interphase [23]. Elevated WDR82 and/or H3K4me3 is associated with therapeutic sensitivity in breast, cervical, and ovarian cancers as well as glioblastomas [24,25,26]. Recent studies show that WDR82 expression is associated with therapeutic sensitivity to platinum drugs [24,27]. However, its role in pediatric gliomas has not been investigated.

In this study, H3K4me3 levels and modifiers for this epigenetic mark were mapped using in-house pediatric glioma specimens and via in silico analysis of multiple publicly available databases. WDR82 and H3K4me3 were found to be elevated in pHGGs and associated with reduced chemotherapeutic sensitivity. A reduction in H3K4me3 by the downregulation of WDR82 decreased H3K4me3 promoter occupancy and gene expression and increased the response of pediatric glioma cells to chemotherapy. Our data suggest that H3K4me3 status is an important determinant of pediatric glioma malignancy and therapeutic response, and that WDR82, which regulates H3K4me3, is a promising target to increase therapeutic efficacy and improve the prognosis for children with malignant gliomas.

## 2. Materials and Methods

### 2.1. Clinical Specimens and Immunohistochemistry (IHC)

Formalin-fixed, paraffin-embedded (FFPE) pediatric glioma samples were used for immunohistochemical analysis. Samples were collected from patients diagnosed in the Department of Pathology, Ann & Robert H. Lurie Children’s Hospital of Chicago (A&RHLCH, Chicago, IL, USA) under IRB# 2005–12252 (Principal investigator: Dr. Tadanori Tomita). Clinicopathological information is summarized in Appendix A. All tumor samples were reviewed by a senior pediatric pathologist (N.W), using World Health Organization 2016 criteria for tumor classification. IHC was performed on FFPE slides using antibodies against H3K4me3 (Cell Signaling Technology, #9727, 1:200) as per manufacturer’s instructions. Images were captured on a Leica DMR-HC upright microscope (Leica Microsystem Inc., Buffalo Groove, IL, USA) and analyzed using OpenLab 5.0 software (PerkinElmer, Waltham, MA, USA). H3K4me3-positive staining was graded semi-quantitatively on a five-tier scale: 0 < 10%, 1+ = 10–25%, 2+ = 25–50%, 3+ = 50–75%, 4+ ≥ 75% for positive tumor cell nuclei.

### 2.2. Cell Lines and Cultures

Pediatric SJ-GBM2 glioblastoma cells were obtained from the Children’s Oncology Group Cell Culture and Xenograft Repository. Pediatric KNS42 cells were purchased from the JCRB (Japanese Cancer Research Resources) cell bank. The human H3K27M DIPG cell line SF8628 was generously provided by Dr. Rintaro Hashizume (Department of Pediatrics, Lurie Children’s Hospital of Chicago). All cells were propagated as monolayers in complete medium consisting of Dulbecco’s modified Eagle’s medium (DMEM, Cat#11965-092) supplemented with 10% fetal bovine serum (FBS, Cat#10082147) from Gibco (Thermo Fisher Scientific, Waltham, MA, USA), at 37 °C with 5% CO_2_.

### 2.3. Plasmids

A set of SMARTvectors inducible lentiviral shRNA plasmids with hEF1 promoter and TurboGFP reporter against human WDR82 (Entrez Gene 80335) V3SH11252-229426599 (clone id:V3IHSHEG_9364249, shRNA#1), V3SH11252-2289952521 (clone id:V3IHSHEG_8890171, shRNA#2), and non-targeting SMART vector inducible lentiviral with hEF1 promoter and TurboGFP reporter control plasmid (Cat#VSC6572) were purchased from Horizon Discovery Ltd. (Cambridge, UK). These vectors were amplified and purified as per manufacturer’s instructions.

### 2.4. Plasmid Transduction

Tumor cells were plated a day before transduction, in 6-well plates at 30–40% confluency, to avoid cell clumps and uneven distribution that can reduce the efficacy of viral transduction. Individual wells of confluent cells were treated with packaged lentivirus (100–500 µL) and polybrene 4–8 µg/mL, followed by incubation for 6–14 h. Cell viability and protein expression were monitored. The transduction process was repeated until an optimal amount of GFP protein was expressed. GFP protein expression was checked after 48–72 h using a fluorescent microscope to visualize the fluorescent tag present in the construct. Cells were then selected with antibiotics. In cases where the construct needed a substrate such as doxycycline (Dox) for induction, Dox was added and protein expression was monitored after 24 h, 48 h, and 72 h.

### 2.5. Total Nuclear and Histone Protein Extraction and Immunoblotting

Total proteins were extracted with Tissue Extraction Buffer I (Life Technologies, cat# FNN0071) with proteinase (Cell Signaling Technology, Beverly, MA, USA), phosphatase (Sigma, St. Louis, MO, USA) inhibitor cocktails, and phenylmethylsulphonyl fluoride (PMSF, Roche, Indianapolis, IN, USA). Total histone was extracted using histone extraction kit (ab113476, Abcam) as per manufacturer’s instructions. Protein concentrations were quantified with the BCA Protein Assay Kit (Thermo Fisher Scientific Inc. Waltham, MA, USA) using Nanodrop ND-2000 (Thermo Fisher Scientific Inc. Waltham, MA, USA). Equal amounts of cell lysate were resolved by sodium dodecyl sulfate–polyacrylamide gel electrophoresis and transferred to nitrocellulose membranes (Bio-Rad, Hercules, CA, USA). Blocking was performed for 60 min with 5% nonfat dry milk in Tris-buffered saline and Tween 20, followed by blotting with primary antibodies overnight at 4 °C. Primary antibodies include polyclonal anti-rabbit WDR82 (Cat#99715, 1:1000), H3K4me3 (Cat#9727, 1:1000), and H3 (Cat#9715, 1:2500) from Cell Signaling Technology (Danvers, MA, USA); β-actin (ab8227,1:3000) from Abcam (Waltham, MA, USA); hSETD1A from Bethyl Laboratories (Cat#A300-289A, 1:1000, Montgomery, TX, USA)); rabbit polyclonal anti-GAPDH (sc-25,778, 1:2000) from Santa Cruz Biotechnology. After washing with Tris-buffered saline and Tween 20, membranes were incubated for 1 h at room temperature with horseradish peroxidase (HRP) conjugated donkey anti-rabbit antibody (sc-2305, 1:5000), and signal was detected with enhanced chemiluminescence substrate (Bio-Rad Laboratories).

### 2.6. In Silico Public Dataset Analysis

Two expression profiling datasets (GSE50161, GSE73038, GSE68015, and GSE36245) were downloaded from the GEO database at the NCBI. Original gene expression profiles of glioma were obtained from these datasets. Expression of individual genes was identified with GEO2R. Adult low- and high-grade glioma data in TCGA were processed with the Gliovis online portal (http://gliovis.bioinfo.cnio.es/, accessed on 15 March 2022). Data were further analyzed with GraphPad Prism 9.0 for gene expression, gene expression correlation, and survival analysis (GraphPad Software, Inc. La Jolla, CA, USA).

### 2.7. RNA-seq

Total RNA was prepared using the RNeasy Mini Kit (Qiagen, Germantown, MD, USA. Cat#74106) as per manufacturer’s instructions. Stranded mRNA-seq was conducted in the Northwestern University Sequence (NUseq) Core Facility. Briefly, total RNA samples were checked for quality using RINs generated from Agilent Bioanalyzer 2100 (Agilent Technologies, Santa Clara, CA, USA). RNA quantity was determined with the Qubit fluorometer (Qubit 2.0, Thermofisher, Waltham, MA, USA). The Illumina TruSeq Stranded mRNA Library Preparation Kit (Cat#20020595, Illumina, Inc., San Diego, CA, USA) was used to prepare sequencing libraries from 1 mg of high-quality RNA samples (RIN = 10). The Kit procedure was performed without modification. The procedure includes mRNA purification and fragmentation, cDNA synthesis, 3′ end adenylation, Illumina adapter ligation, and library PCR amplification and validation. Illumina HiSeq 4000 sequencer (San Diego, CA, USA) was used to sequence the libraries with the production of single-end, 50 bp reads at the depth of 30-40 M reads per sample. The quality of reads, in FASTQ format, was evaluated using FastQC. Adapters were trimmed and reads of poor quality or aligning to rRNA sequences were filtered. The cleaned reads were aligned to the H. sapiens genome using STAR [28]. Read counts for each gene were calculated using HTseq-count, in conjunction with a gene annotation file for hg38 obtained from UCSC (http://genome.ucsc.edu, accessed on 14 March 2021). Differential expression was determined using DESeq2. The cutoff for determining genes, which showed significant differential expression, was an FDR-adjusted *p*-value less than 0.05.

### 2.8. ChIP-seq

Fragmental DNA for ChIP-seq was prepared using SimpleChIP^®^ Plus Enzymatic Chromatin IP Kit (Magnetic Beads) (Cell Signaling Technologies, Danvers, MA, USA, Cat#9005) per manufacturer’s instructions. ChIP-seq was conducted in the NUseq Core Facility. Briefly, ChIP and input DNA samples were checked for quality and quantity using Qubit and Agilent Bioanalyzer (Agilent Technologies, Santa Clara, CA, USA). After passing quality control), the Illumina TruSeq ChIP Sample Preparation Kit (IP-202-1012 and IP-202-1024, Illumina, Inc., San Diego, CA, USA) was used to prepare sequencing libraries from 5-10 ng of ChIP and input DNA. This procedure includes end repair, 3-end adenylation, Illumina adapter ligation, and library PCR amplification and validation. Illumina HiSeq 4000 sequencer was used to sequence the libraries with the production of single-end, 50 bp reads at the depth of 30–40 M reads per sample. After sequencing, the quality of reads, in FASTQ format, will be evaluated using FastQC. Adapters will be trimmed; reads of poor quality will be filtered. The cleaned reads will be aligned to the H. sapiens genome (hg38) using Bowtie [29]. Peak calling and differential peak analysis will be performed using HOMER (http://homer.ucsd.edu/homer/index.html, accessed on 26 January 2021).

### 2.9. Cell Proliferation Assay

SJ-GBM2 and SF8628 cells were used to determine if reducing WDR82 through inducible knockdown affected cell viability; 1 × 10^4^ cells/100μL were plated in 96-well plates with complete cell culture medium with or without Dox (2 µg/mL), and subjected to 3-(4, 5-dimethylthiazol-2-yl)-5-(3-carboxymethoxyphenyl)-2-(4-sulfophenyl)-2H-tetrazolium (MTS, Promega, Madison, WI, USA) assay. Pediatric high-grade glioma KNS42 cells were seeded into 6-well tissue culture plates and allowed to adhere. Attached cells were treated with or without Dox (SKU#D9891-10G, Sigma-Aldrich) at 2 µg/mL. Cells were incubated with Dox for 2 weeks, at which time, colonies were counted following staining with methylene blue (0.66% solution in 95% ethanol). Plating efficiencies were calculated as the ratio of the number of colonies formed to the number of cells seeded. The cell lines were used for different assays due to their distinct growth patterns.

### 2.10. In Vivo Studies

Briefly, 6–8-week-old athymic nude mice were purchased from Taconic. All mice were housed under aseptic conditions, which included filtered air and sterilized food, water, bedding, and cages. The Institutional Animal Care and Use Committee (IACUC) approved all animal protocols. SJ-GBM2 (1 × 10^5^) or KNS42 (2 × 10^5^) cells, transduced with sh-WDR82 vectors, or non-transduced controls, were implanted into the right striatum. Mice were given 1% sucrose or 1% sucrose + 2 mg/mL Dox in their drinking water [30], which was replaced every 2–3 days. Animals were monitored daily and body weight was measured every 2–3 days. Mice were euthanized with CO_2_ asphyxiation followed by cervical dislocation when they became moribund (e.g., >20% weight loss, neurologic symptoms, or evidence of pain/distress). Brains were harvested and fixed with 10% paraformaldehyde in PBS overnight and switched to PBS prior to embedding. The tissue was sectioned onto slides and stained with H&E. Tissue preparation was conducted at the Mouse Histology and Phenotyping Laboratory (MHPL), Northwestern University Feinberg School of Medicine.

### 2.11. Statistical Analysis

Cell proliferation results were read on a Synergy 2 Microplate Reader (BioTek Instruments Inc., Winooski, VT, USA). Cell survival is presented as a percentage of viable cells compared to the viable cell number in the corresponding control, with the control set as 100%. P values were calculated using two-way ANOVA, with *p* < 0.05 considered significant. Statistical tests were 2-sided. Kaplan–Meier and *t*-tests were performed to compare survival between groups. Graph generation and statistical analyses were performed with GraphPad Prism 9 software (GraphPad Software, Inc. La Jolla, CA, USA).

## 3. Results

### 3.1. H3K4me3 Levels Increase with Histopathological Malignancy in Pediatric Gliomas

To investigate whether a correlation exists between clinicopathologic variables and H3K4me3, forty-six pediatric low-grade gliomas (WHO grades I and II) and ten pediatric high-grade gliomas (WHO grades III and IV) FFPE specimens were IHC-stained for H3K4me3 (Appendix A). H3K4me3 was predominantly observed in cell nuclei with the frequency of immunopositive cells ranging from 0% to 100%. Patient gender and tumor location are not associated with H3K4me3 levels. Patient age and WHO grade positively correlated with H3K4me3 levels. IHC results showed H3K4me3 significantly increased in pHGGs, relative to pLGGs (Figure 1A,B). Multivariate Cox proportional hazards analysis showed that progression-free survival of patients with high levels of H3K4me3 (IHC score ≥ 3) was significantly shorter relative to patients with low levels of H3K4me3 (IHC score ≤ 2) (Figure 1C). Western blots of protein extracts from tissue samples confirmed higher levels of H3K4me3 in pHGGs vs. pLGGs (Figure 1D). These results indicate that H3K4me3 IHC staining is a potential predictor of pediatric glioma malignancy, especially in pHGGs, irrespective of the presence or absence of histone H3 mutations.

### 3.2. WDR82 Plays a Distinct Role in the Tumor Biology of Pediatric Glioma

H3K4 is reversibly modified by methyltransferases and demethylases. To identify the key modifier(s) of H3K4 in pHGGs, gene expression profiling with microarray analysis revealed that WDR82, among all H3K4 modifiers, is differentially expressed at significant levels and correlates with WHO-grade malignancy, regardless of histone mutation subtypes (Figure 2A,B). These results are supported by in silico analysis of a glioma gene expression profiling dataset (GSE50161, Appendix A). Geneontology function analysis indicates that WDR82 expression is related to multiple biological processes, molecular functions, and cellular signaling pathways (Figure 2C). The level of WDR82 shows an inverse correlation with patient survival (Figure 2D). Interestingly, in silico analysis results from The Cancer Genome Atlas (TCGA) show that WDR82 does not correlate with WHO-grade malignancy and survival in adult gliomas (Figure 2E,F). In combination, these results indicate that WDR82 expression is specifically relevant to the tumor biology of pHGGs.

Altered gene expression following WDR82 inducible knockdown. We have shown that WDR82 is highly expressed in pHGGs vs. pLGGs. To investigate the role of WDR82 in pHGGs, pHGG cell lines SJ-GBM2, SF8628, and KNS42 were transduced with SMARTvector inducible lentiviral shRNA targeting WDR82 (shRNA#1 and shRNA#2) or non-targeting control vector (Appendix A). The results show that GFP is expressed (green fluorescence) at day 5 in transduced cells following Dox (2 µg/mL) treatment, whereas untreated cells show no fluorescence (Appendix A). We further checked WDR82 expression in these cells by real-time PCR (Appendix A) and Western blot (Appendix A). WDR82 level significantly decreased.

To identify altered gene expression, with WDR82 knockdown, total RNA was extracted from transduced cells in the absence or presence of Dox (2 µg/mL) at day 5 and sequenced (RNA-seq and ChIP-seq with H3K4me3 antibody immunoprecipitation). The RNA-seq results from SJ-GBM2 cells indicate that 129 genes were differentially (cutoff: 1.5 times) expressed in cells transduced with shRNA#1 and shRNA#2, in comparison to non-significant gene expression changes with non-target shRNA (shNT) controls in the presence or absence of Dox (Figure 3A–C). We further analyzed the function of these genes using the Gene Ontology online portal (http://geneontology.org/, accessed on 15 March 2021). The results showed that reducing WDR82 significantly decreases the expression of genes associated with mitosis, proliferation, and DNA repair (Figure 3D). ChIP-seq results indicate that LIN9 and NUP43 promoter H3K4me3 levels decreased in SJ-GBM2 cells transduced with inducible shRNA#2 against WDR82 following Dox induction (Figure 3E).

### 3.3. WDR82-Mediated H3K4me3 Is Associated with Mitotic and Cell-Cycle-Related Gene Regulation

To investigate if changes in gene expression link to WDR82-mediated H3K4me3 alteration at their promoters, representative genes were selected. The correlation between gene expression and WDR82 was plotted using RNA-seq results from pediatric glioma tissue specimens. Promoter H3K4me3 levels were mapped using ChIP-seq results SJ-GBM2 cells treated with the Dox inducible shRNA vector against WDR82 (shRNA#2), in the absence and presence of Dox. The representative results showed WDR82 did not correlate with the expression of NUP62 and FBOX10, genes for which levels increased in pediatric glioma tissue specimens (Figure 4A). Additionally, H3K4me3 at their promoters was unaltered (Figure 4B). In the group of genes with decreased expression, there was a correlation with WDR82 in pediatric glioma tissue specimens, and representative results are shown for NUP43 and PKYMT1 (Figure 4C). Furthermore, H3K4me3 promoter levels were undetectable in SJ-GBM2 cells treated with inducible shRNA#2 against WDR82, following Dox or non-Dox treatment (Figure 4D).

### 3.4. WDR82 Knockdown Decreases In Vitro Cell Viability and DNA Damage Repair in pHGG Cells

To investigate whether WDR82 levels affect pHGG cell growth, equal numbers of cells were plated in 96-well plates in the absence and presence of Dox (2 µg/mL), with relative cell numbers compared at 0~7 days by MTS assay. OD values for Dox-treated groups were normalized based on the values of corresponding untreated cells. Inducible knockdown of WDR82, following Dox treatment, significantly suppresses cell growth in SJ-GBM2 and SF8628 cells (Figure 5A). Inducible knockdown of WDR82 also affects pHGG KNS42 cell proliferation, as indicated by colony formation assay (Figure 5B,C). KNS42 cells were plated (125, 250, and 500 cells) in 6-well plates and cultured in the absence or presence of Dox (2 µg/mL) for two weeks prior to counting colonies stained with methylene blue (0.66% solution in 95% ethanol).

CDK2 is associated with CCND1, which regulates cell proliferation, the cell cycle, and DNA damage repair in childhood cancer through the regulation of Rb [31]. CDK2 decreased in SJ-GBM2 cells treated with inducible shRNA#2 against WDR82, following Dox vs. non-Dox treatment (Figure 3C). We hypothesized that CDK2 and CCND1 are regulated by the WDR82 mediation of promoter H3K4me3. To verify this hypothesis, CDK2, CCND1, and WDR82 expressions were examined in pediatric gliomas. A significant correlation was identified (Figure 5D). ChIP-seq results showed that at the promoters of these two genes, H3K4me3 decreased in SJ-GBM2 cells, treated with inducible shRNA#2 against WDR82, following Dox vs. non-Dox treatment (Figure 5E). These results indicate that WDR82-mediated H3K4me3 alteration is associated with cell proliferation, the cell cycle, and DNA damage repair (Figure 5F).

### 3.5. WDR82 Knockdown Increases Therapeutic Sensitivity against DNA-Damaging Agents and Radiation Sensitivity

Increased WDR82 expression is associated with tumor malignancy grade (Figure 2A). Recent studies have shown that the overexpression of WDR82 promotes high H3K4me3 in tumors following chemotherapy [24,27]. Our RNA-seq results from SJ-GBM2 cells indicate that reduction of WDR82 impairs DNA repair and increases apoptosis. We hypothesized that reduction of WDR82 could increase sensitivity to chemotherapeutic agents that break double-stranded DNA, and to radiation. To verify this hypothesis, the effect of inducible WDR82 knockdown was investigated in SJ-GBM2 and SF8628 cells treated with cisplatin (CDDP), a first-line clinical drug for treating pHGG patients. Inducible WDR82 knockdown by Dox increased sensitivity to CDDP (Appendix A). We also tested whether inducible WDR82 knockdown by Dox increases cell response to radiation following a clonogenic assay protocol as described [32]. WDR82 reduction increased radiation sensitivity in KNS42 cells, as indicated by a comparison of colony numbers (Appendix A) and calculating the dose of enhancement factors (DEFs) at 10% survival (Appendix A).

### 3.6. WDR82 Knockdown Decreases In Vivo pHGG Tumor Growth and Extends Survival of pHGG-Tumor-Bearing Mice

WDR82 contributes to tumorigenesis, malignant phenotype, and tumor proliferation of multiple human cancers [33,34]. To investigate if WDR82 and H3K4me3 correlate to proliferation in pediatric glioma specimens, PCNA, MIK67, WDR82, and H3K4me3 expressions were examined. WDR82 correlated with PCNA and MIK67 (Figure 6A) and Ki67 was associated with H3K4me3 (Figure 6B). To investigate if WDR82 knockdown decreases pHGG tumor growth in vivo, 1 × 10^5^ cells (SJ-GBM2 transduced with shNT and WDR82 shRNA#2) were inoculated intracranially into the right striatum of 6–8-week-old athymic nude mice. Mouse housing and tissue processing as described under in vivo studies. Tumor size and mitotic cells are not significantly different in non-targeted shRNA (shNT) groups with or without Dox. In contrast, tumor size is smaller and there are fewer mitotic cells in the group treated with Dox, as indicated by comparing results from animals given Dox vs. 1% sucrose, (Figure 6C–E). The results also showed that survival was significantly extended in mice in the WDR82 shRNA#2 + Dox group (Figure 6F). These results indicate that suppression of WDR82 decreases cell growth and extends the survival of tumor-bearing animals.

## 4. Discussion

In a previous study, we established that H3K4me3 is associated with WHO-grade malignancy in pediatric ependymomas [14]. In this work, we found that H3K4me3 is also associated with WHO grade in pediatric gliomas, with higher levels of H3K4me3 in HGGs vs. LGGs (Figure 1A). Patients with higher tumor levels of H3K4me3 had a shorter progression-free survival (Figure 1C). Furthermore, WDR82, an H3K4me3 methyltransferase, is elevated in pHGGs vs. LGGs, which is inversely correlated with clinical prognosis (Figure 2). Furthermore, we also found that WDR82-meditated H3K4me3 alters gene expression related to a variety of biological functions including stem cell features, cell proliferation, the cell cycle, and DNA damage repair. The results highlight the necessity of a thorough understanding of the biological roles of WDR82 and H3K4me3 in pediatric glioma. This knowledge will establish if WDR82 and H3K4me3 are potential epigenetic targets to suppress tumor progression, increase therapeutic efficacy, and improve outcomes for children with malignant gliomas.

The discovery of recurrent histone mutations in children with pHGGs in the last decade revealed that PTMs on histone tails are one of the milestone changes whose role must be elucidated in cancer research. In pediatric gliomas, H3G34V/R in hemisphere tumors and H3K27M in midline gliomas including DIPG are common histone mutations [2,3]. These mutations crosstalk with other histone modifications and bring about abnormal gene expression involved in pediatric malignant glioma tumorigenesis. Recently, H3K36me2 and H4K16Ac were discovered to be novel epigenetic signatures of DIPG [35]. H3K4me3, an important histone mark for developmental neurogenesis, was relatively unaffected regardless of these mutations as shown by Western blot analysis [8,9,10]. However, ChIP-seq analysis to investigate promoter H3K27me3 and H3K4me3 in wild-type (WT) and H3K27M NSCs and DIPGs showed elevated promoter H3K4me3 in H3K27M-induced genes such as Lin28b, Igf2bp2, Plag1, Pbx3, Eya1, etc. These genes regulate neuroprogenitor cell proliferation and differentiation and are associated with the DIPG oncogenic signature [36]. Another study used ChIP-seq to map promoter H3K27me3 and H3K4me3 in DIPGs with and without H3K27M knockdown, promoter H3K4me3 changed in differentially expressed genes, with increased levels in K27M-induced upregulated genes, for example, RORB and VIM [37]. H3K4me3 is activated in pHGGs with an H3.3G34R/V mutation in comparison with WT tumors [38]. Pediatric diffuse midline gliomas with an H3K27M mutation have lower levels of methionine adenosyltransferase 2A (MAT2A) protein; depletion of residual MAT2A reduces global H3K4me3 [39]. In this study, we determined that H3K4me3 is elevated in pHGGs vs. pLGGs (Figure 1 and Appendix A), and its reduction via decreasing WDR82 alters the expression of genes involved in cell proliferation, the cell cycle, and DNA damage repair. Altogether, these findings suggest H3K4me3 impacts pHGGs, which merits further investigation.

Among the six methyltransferases for H3K4me3, SETD1A is associated with WHO malignancy in pediatric gliomas (Figure 1D and Appendix A). Investigation of its subunits WDR82, ASH2L DPY30 CXXC1, RBBP5, and WDR5 showed that WDR82 and WDR5 were significantly associated with WHO-grade malignancy in pediatric gliomas (Figure 1C, Figure 2 and Appendix A). WDR5 is also associated with malignancy in adult gliomas [40]. WDR82 is relatively high in glioma vs. other cancers (Appendix A), which is the sole subunit in SETD1A-COMPASS, specifically associated with pediatric glioma WHO-grade malignancy (Figure 2A) and patient survival (Figure 2D), regardless of histone mutation status. It is not correlated with adult glioma WHO-grade malignancy or patient survival (Figure 2E,F). Due to this specificity for pediatric gliomas, WDR82-mediated H3K4me3 is the focus of this study.

WDR82 activity is a key factor in maintaining embryonic [34] and cancer stem cells [41,42,43]. In this work, sphere formation was impaired in shWDR82-transduced cells following Dox treatment, in comparison to non-Dox treatment and shNT groups, respectively (Appendix A). Results from in silico analysis of dataset GSE50161 showed a positive correlation between WDR82 and the expression of SOX2, (Appendix A), a glioma stem cell determinant [41], and one of the genes decreased in Figure 2B. ChIP-seq results from SJ-GBM2 cells show that WDR82 knockdown causes promoter H3K4me3 reduction at the SOX2 gene (Appendix A). These results are consistent with previous findings [22,42] and provide further confirmation that WDR82-mediated H3K4me3 is associated with stem cell characteristics.

In the present study, WDR82-mediated H3K4me3 alters the gene expression of NUP43 [43], LIN9 [44], PKMYT1 [45], CDK2 [46], LOX [47], and NUP62 [48], genes associated with mitosis, the cell cycle, and DNA damage repair in pHGGs (Figure 3, Figure 4 and Figure 5). WDR82 association with these genes is a novel finding; however, the results are consistent with discoveries in embryonic stem cells and various human cancer cells. For instance, WDR82-mediated H3K4me3 is responsible for facilitating M-phase progression in mixed-lineage leukemia. WDR82 knockdown increases mitotic cells [49]. WDR82 modulates cell cycle progression through the regulation of the B-cell translocation gene 2 (BTG2). Depletion of WDR82 induces the expression of BTG2, an anti-proliferative protein [50]. WDR82-mediated H3K4me3 reduction induces apoptosis in embryo stem cells [22]. WDR82 and/or H3K4me3 are associated with chemotherapeutic sensitivity in breast, cervical, and ovarian cancers, and adult glioblastoma cells [24,25,26]. WDR82 is an important binding partner with TOX4 in HeLa cells following cisplatin treatment [24,27]. Our results, showing that inducible reduction of WDR82 sensitizes the pediatric glioma cell response to cisplatin and radiation therapy (Appendix A), are also in line with alterations in gene expression observed in this work. Taken together these findings demonstrate that WDR82 regulates the gene expression associated with mitosis, proliferation, and the cell cycle, as well as apoptosis and chemotherapeutic response in pHGGs, and thus may be a potential therapeutic target.

## 5. Conclusions

In summary, H3K4me3 and WDR82 are elevated in pHGGs vs. pLGGs. WDR82-mediated H3K4me3 is associated with the expression of genes involved in regulating stem cell features, cell mitosis and proliferation, the cell cycle, and DNA damage repair. Reduction of WDR82 increased the response of pediatric glioma cells to chemotherapy. Inducible reduction of WDR82 decreases pHGG tumor cell growth in vivo and extends animal survival. These findings suggest that WDR82-mediated H3K4me3 is a significant factor in pediatric glioma, and further investigation of WDR82 as a promising epigenetic therapeutic target for pHGG is warranted.

## Figures and Tables

**Figure 1 cancers-15-03429-f001:**
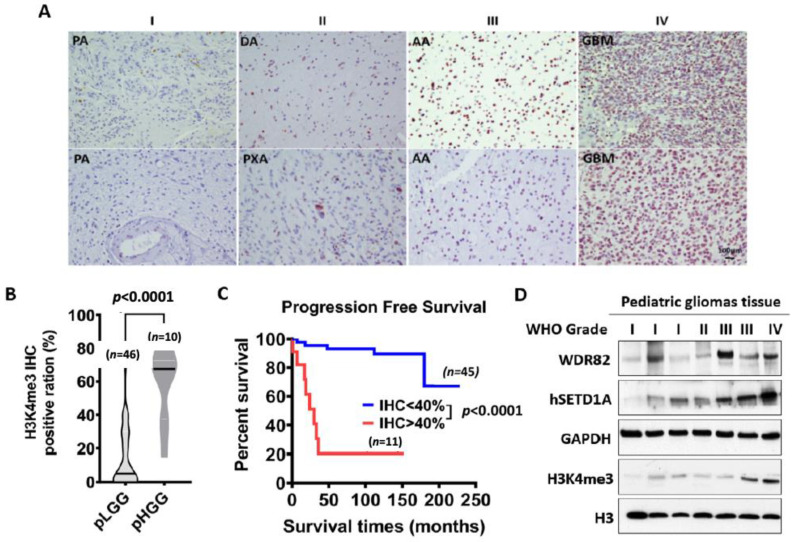
H3K4me3 is associated with malignancy and prognosis of pediatric gliomas: (**A**). Immunohistochemistry (IHC) shows H3K4me3 in WHO-grade I–IV pediatric gliomas. (**B**). Quantitative IHC results for H3K4me3 in pediatric gliomas. (**C**). Survival analysis of pediatric gliomas based on H3K4me3 IHC results. (**D**). Expressions of WDR82, SETD1A, and H3K4me3 detected with Western blots in fresh pediatric glioma specimens. Abbreviations: PA, pilocytic astrocytoma; DA, diffuse astrocytoma; PXA, pleomorphic xanthoastrocytoma; AA, anaplastic astrocytoma; GBM, glioblastoma.

**Figure 2 cancers-15-03429-f002:**
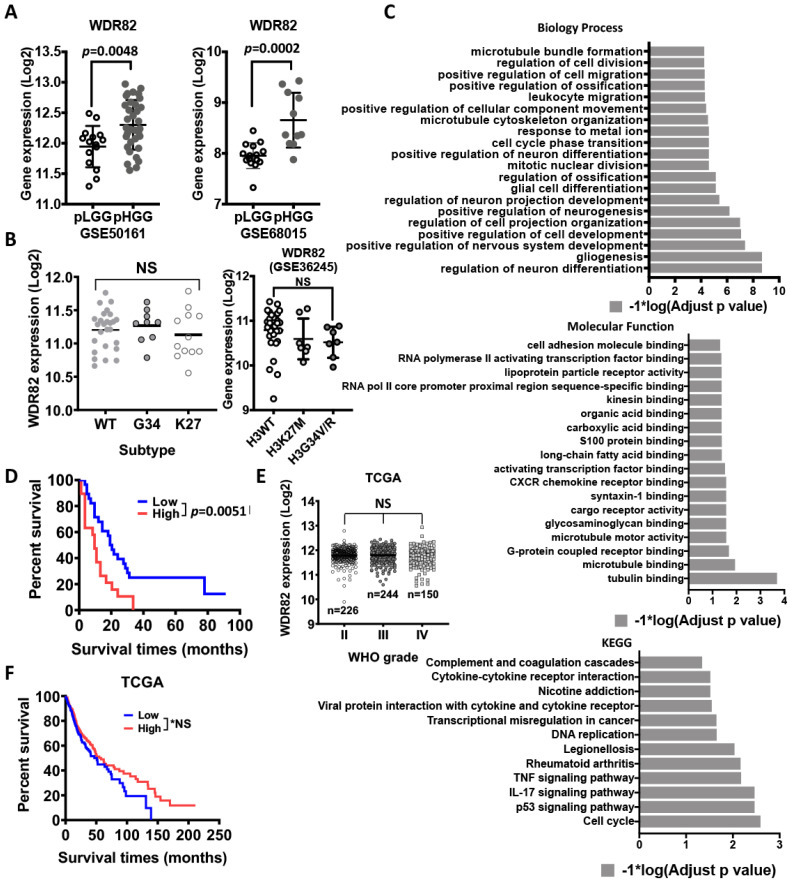
WDR82 is associated with malignancy in pediatric gliomas: (**A**). WDR82 expression in WHO low- and high-grade pediatric gliomas. (**B**). WDR82 expression in molecular subtypes of pediatric high-grade gliomas (pHGGs). (**C**). In silico analysis shows WDR82 function in pediatric gliomas. (**D**). Survival analysis of pHGGs based on WDR82 expression levels. (**E**,**F**. NS: no significance). In silico analysis of TCGA database shows WDR82 expression in adult WHO-grade gliomas and survival analysis of adult gliomas based on WDR82 expression. * *p* < 0.05.

**Figure 3 cancers-15-03429-f003:**
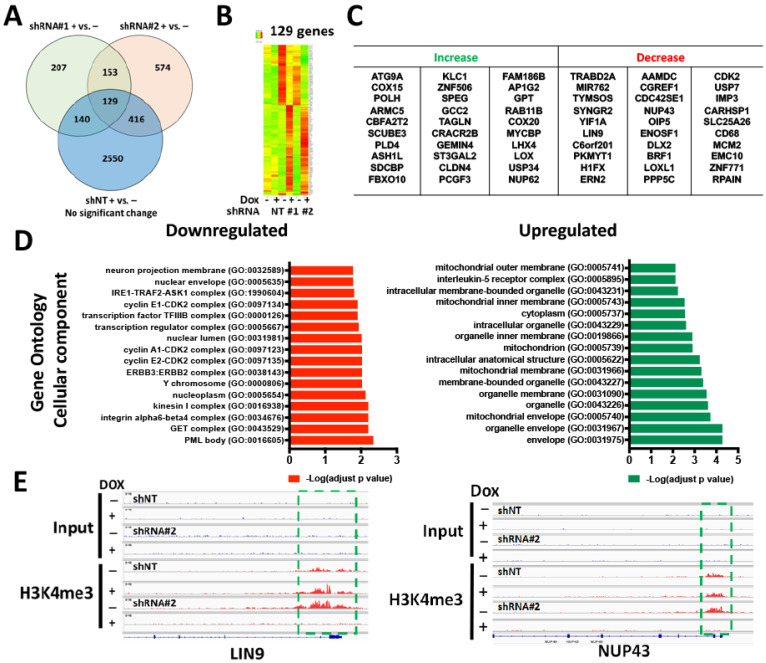
Altered gene expression following WDR82 inducible knockdown in pediatric SJ-GBM2 glioma cells: (**A**). Venn diagram shows differentially expressed genes from RNA-seq results, generated from total RNA extracted from shWDR82-transduced SJ-GBM2 cells in the presence (+) and absence (−) of doxycycline. (**B**,**C**). The heat map showed 129 differential expressed genes (**B**) and the top 30 genes for which expression increased or decreased in (**A**). (**D**). Gene Ontology Molecular function analysis for the top-down or upregulated genes and their associated functions. (**E**). An example from ChIP-seq results shows decreased promoter H3K4me3 levels in the LIN9 and NUP43 genes following Dox induction.

**Figure 4 cancers-15-03429-f004:**
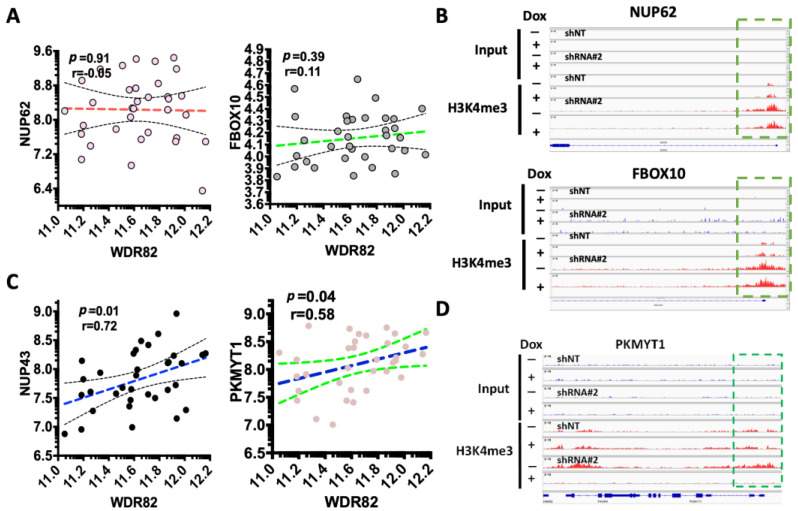
WDR82 regulates expression of mitotic and DNA damage repair genes through mediation of H3K4me3: (**A**,**B**). Representative dot plots show genes that are not associated with WDR82 in pediatric glioma tissue specimens (**A**); for these genes, H3K4me3 levels at the promoter are not altered as shown by ChIP-seq results (**B**) from SJ-GBM2 cells, treated with Dox inducible shRNA against WDR82, in the absence and presence of Dox. (**C**,**D**). Representative dot plots showing genes that are associated with WDR82 in pediatric glioma tissue specimens (**C**) for these genes; i.e., PKMYTI, H3K4me3 promoter levels decreased as shown with ChIP-seq results (**D**) from SJ-GBM2 cells, treated with Dox inducible shRNA against WDR82, in the absence and presence of Dox.

**Figure 5 cancers-15-03429-f005:**
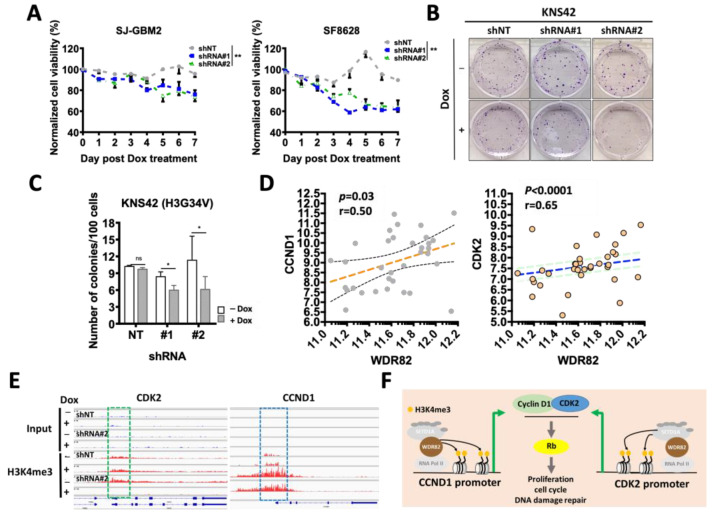
WDR82 knockdown decreases in vitro cell viability and proliferation and extends survival of pHGG-tumor-bearing mice: (**A**). Normalized cell viability of SJ-GBM2 and SF8628 cells transduced with inducible shRNA #1 or #2 against WDR82 or non-target shRNA (shNT) in the absence or presence of Dox (2 µg/mL), in comparison to parental non-treated cells. (**B**,**C**). Representative images (**B**) and quantitative results (**C**) showed colony formation in 500 KNS42 cells transduced with shRNAs #1 or #2, or shNT in the absence or presence of Dox (2 µg/mL). (**D**). Correlations between WDR82 and CDK2 in pediatric gliomas. (**E**). ChIPseq results for H3K4me3 levels at CDK2 and CCND1 promoters in the absence and presence of Dox in SJ-GBM2 cells. (**F**). The illustration indicates that WDR82-mediated H3K4me3 at CCND1 promoter regulates cell proliferation, cell cycle, and DNA damage through activating the cyclin D1-CDK2 signaling pathway. (* *p* < 0.05; ** *p* < 0.01).

**Figure 6 cancers-15-03429-f006:**
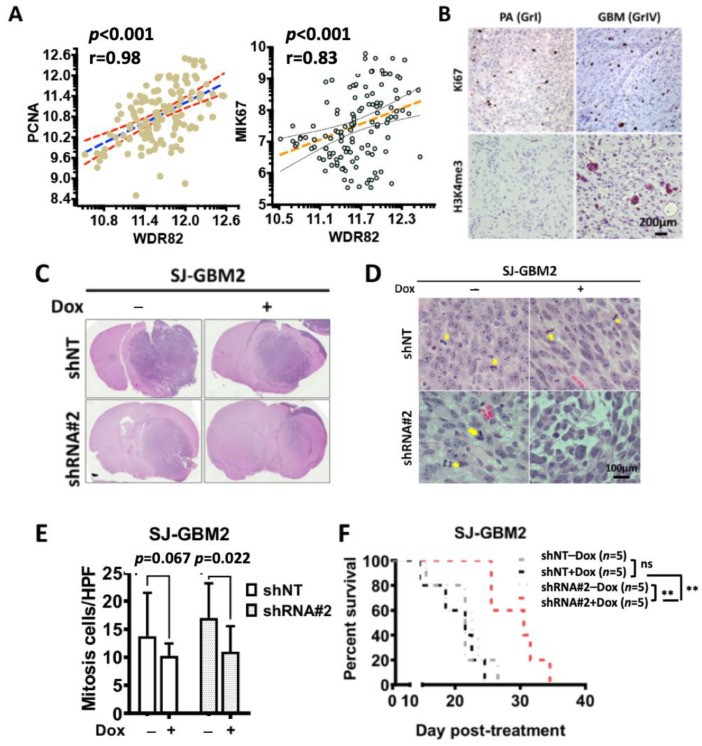
WDR82 and H3K4me3 are associated with cell proliferation; WDR82 knockdown decreases in vivo pHGG tumor growth and extends survival of pHGG-tumor-bearing mice: (**A**). Correlation between PCNA, MIK67, and WDR82 in pediatric glioma tissue specimens. (**B**). Representative immunohistochemistry staining shows Ki67 and H3K4me3 in pediatric astrocytoma (PA, WHO grade I, GrI) and glioblastoma (GBM, WHO grade IV, GrIV) tumor samples. (**C**). Representative HE images show tumors in shNT and shRNA#2 treated cells with or without Dox (2mg/mL). (**D**,**E**). Mitotic cells in HE stained slides (**D**) and relative quantification per high power field (400×) (**E**) from in vivo intracranial SJ-GBM2 xenograft tumors with or without Dox induction. (**F**). Survival curve for mice inoculated with SJ-GBM2 cells transduced with shNT and shRNA#2 against WDR82 with or without Dox. (** *p* < 0.01).

## Data Availability

The RNA- and ChIP-sequence data are available upon request.

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
