# Peer review of "WDR82-Mediated H3K4me3 Is Associated with Tumor Proliferation and Therapeutic Efficacy in Pediatric High-Grade Gliomas"

_cancers, 2023, doi:10.3390/cancers15133429_

Round 1

Reviewer 1 Report

Wadhwani N. et al. investigated the role of H3K4me3  and WDR82 in pediatric high grade gliomas, addressing their oncogenic role and mechanisms. Data regarding the molecular drivers of these tumors surely are of interest considering their dismal outcome and the lack of effective treatments, but some significant issues should be addressed:

- In silico analysis: in the Methods, it is stated that the GSE50161 and the GSE73038 datasets were used, but, especially the second one, is mainly made of embryonal and other tumors rather than gliomas. It can be useful to specify which sample types were used from these datasets. Moreover, additional datasets are reported in the results (Figure 2): GSE68015 and GSE36245.

- Different cell lines have been used for different assays. For example, SJ-GBM2 and SF8628 for cell proliferation and KNS42 for clonogenic survival assay. It would be useful to specify why a specific cell line was selected for a specific assay. Morover, the three cell lines seem to have been derived from three different tumor types: SJ-GBM2, which possibly is a H3-wildtype pHGG; SF8628, a H3 K27M-mutant pHGG and KNS42, a H3 G34V-mutant pHGG. The analysis of a heterogenous set of cell lines can be useful to gather data about WDR82 role in the different pHGG types, but the same experiments should be performed on all of them or anyway the potential implications of analyzing molecularly heterogenous cell lines should addressed/explained.

- Analysis of the retrospective series:

-- According to the methods, the CNS WHO 2016 classification was used for classifying the retrospective series. I understand that classifications are quickly evolving and it can be difficult to account for these changes in the middle of a project, but since the WHO 2021 has now been published since about 1 year and half, and that it specifically addressed the topic of pediatric tumors, I think it would be important to update this part of the study to it. For example, a glioblastoma diagnosis is unexpected in pediatric patients and the terms diffuse and anaplastic astrocytomas have been removed in WHO 2021 and anyway also these diagnoses are unexpected in this setting. I would also add the specific tumor types included in the analysis since both low grade and high grade pediatric gliomas are highly heterogenous.

-- Regarding these results, I would be more cautious when stating that "H3K4me3 IHC staining is an accurate predictor of pediatric glioma malignancy especially in pHGGs" considered the limited sample size (especially of pHGG) and that this finding could be related to the specific molecular characteristics of the analyzed tumor types more than their malignancy. To fully address this question, a comprehensive/representative series of both pLGG and pHGG should be analyzed, but I understand that this is out of the scope of the study.

- A shRNA#3 is listed in the methods, but it seems to be missing in the results. Please check.

Minor issues:

- Lines 109-110 "WDR82 and H3K4me3 were found to be elevated in pHGGs and associated with chemotherapeutic sensitivity.": I would specify that it's associated with reduced chemotherapeutic sensitivity.

- Lines 217-222: past tense should be used since it refers to what has been done.

Reviewer 2 Report

The manuscript entitled “WDR82 mediated H3K4me3 is associated with tumor proliferation and therapeutic efficacy in pediatric high-grade gliomas” by Nitin et al. comprehensively investigates the role of WDR82-mediated H3K4me3 in pediatric high-grade gliomas (pHGGs). The authors found that WDR82-mediated H3K4me3 is elevated in pHGGs compared to pLGGs and is associated with the expression of genes involved in regulating stem cell features, cell proliferation, cell cycle, and DNA damage repair.

Furthermore, the authors demonstrate that the reduction of WDR82-mediated H3K4me3 increases the response of pediatric glioma cells to chemotherapy, decreases pHGG tumor cell growth in vivo, and extends animal survival. These findings suggest that WDR82-mediated H3K4me3 is a significant factor in pediatric glioma, and further investigation of WDR82 as a promising epigenetic therapeutic target for pHGG is warranted.

The paper is well-written and structured, and the experimental methods are sound. The conclusions drawn from the results are well-supported. The study provides important insights into the potential of WDR82 as an epigenetic target to increase therapeutic efficacy and improve the prognosis for children with malignant gliomas.

I recommend this manuscript for publication, as it significantly contributes to pediatric oncology and could lead to new therapeutic strategies for treating pediatric high-grade gliomas.

Reviewer 3 Report

The manuscript is relevant and interesting. The strategy es well established and the results supports the conclusions. However is very difficult to follow since much information is referred as supplementary with out context.

There is a lack of explanations about the reduction of the work. For example, the refseq experiments in line SJ-GBM2 and not in SF8628 or KNS4; why study LIN9 and NUP43 expression instead other proteins enlisted in figure 3; why viability was followed only in two cell lines; why in vivo experiments were done only with SJ-GBM2 cell line, etc.

Figures have poor quality, should be improved mainly 2 and 3.

Original images of westen blot have not explanation, please explain mainly the saturated ones.

Reviewer 4 Report

Authors do not comment Western blot analysis; it seems that there is correspondence between SETD1 and H3K4me3, while there is no correspondence between the levels of WDR82 and H3K4me3. Is it a single blot or a representative blot? I suggest to introduce quantification?

In suppl. Fig 2 there is no correspondence between RNA and protein in SJ-GBM2 cells. Why dox treatment induce WDR82 protein in these cells?

CDK2 is in the list of downregulated genes (Fig 3A) why CCND1 is not included?

In suppl. Fig 3A (x axis) add μ

In suppl. Fig 5 it seems that dox treatment in SJGBM2 and SF8628 shNT increase the number of spheres > 250 μm. Why?

I suggest to revise text for typos.

English language is fine. Only  minor editing is required.

Reviewer 5 Report

The manuscript “WDR82 mediated H3K4me3 is associated with tumor proliferation and

therapeutic efficacy in pediatric high-grade gliomas” is well written and a very good attempt to evaluate WDR82 mediated H3K4me3 an important determinant of pediatric glioma malignancy and therapeutic response. All experiments are well-planned, and the results validate the objective of the study.

Minor issues.

1.      Author should improve the quality of the image uploaded in figure 3,4 &5.

2.      Why Doxorubicin selected for the study? Did author check with chemotherapeutics drug other than doxorubicin.

3.      In Fig 5D suggest WDR82 mediated H3K4me3 alteration is linked with the gene associated with cell cycle but author did not check which phases of cell cycle is affected by WDR82 mediated H3K4me3 alteration

4.      Fig 6F result showed that survival was significantly in the group of mice treated with WDR82 shRNA#2 + Doxo compared to shNT+Doxo, but in case of tumor size there is no significant changes between these two groups? Did the author checked the tumor size during the course of treatment between these two groups?

Round 2

Reviewer 1 Report

I thank the Authors' for their revision and reply which has addressed, at least in part, my queries.